# Study of the Electrical Conductivity Characteristics of Micro and Nano-ZnO/LDPE Composites

**DOI:** 10.3390/molecules27123674

**Published:** 2022-06-08

**Authors:** Guang Yu, Yujia Cheng, Zhuohua Duan

**Affiliations:** Mechanical and Electrical Engineering Institute, University of Electronic Science and Technology of China, Zhongshan Institute, Zhongshan 528400, China; yuguang@hrbust.edu.cn (G.Y.); duanzhuohua@163.com (Z.D.)

**Keywords:** electrical conductivity characteristics, ZnO/LDPE, micro composite, nano composite

## Abstract

Polyethylene, a thermoplastic resin made by ethylene polymerization, is widely used in electrical insulation. In this study, low-density polyethylene (LDPE) is used as a matrix with micro- and nano-ZnO particles as a filler to produce different proportions of micro- and nano-ZnO composites by melt blending. These samples are characterized by Polarized Light Microscopy (PLM) and FTIR tests, with their conductance measured under different field strengths. The current density vs. electric field strength (J-E) curve of micro- and nano-ZnO composites under different field strengths are measured and analyzed. The J-E curves of different composites at different temperatures are measured to explore conductance with temperature. The results of these tests showed that nano-ZnO composites successfully suppressed conductivity at elevated temperatures and electric field strengths, while micro-ZnO composites increased the conductivity relative to pure LDPE.

## 1. Introduction

With the fast development of science and technology, the requirement for polymer insulating materials continues to grow increasingly higher [1,2,3]. In 1959, the concept of nanotechnology was proposed by Richard Feynman. The nano-technological field has grown rapidly ever since, with nanocomposites also being invented [4,5,6]. The nanocomposites are composites comprising a matrix filled with nanoparticles [7,8]. From existing research, adding a certain content of nanoparticles clearly improves the composite’s electrical properties; these new composites meet the needs of insulation [9,10,11]. In comparison with micro-composites, nanocomposites have less specific surface area, but the interaction between interfaces is stronger [12,13,14,15]. When the material properties change, the content of nanoparticles decreases [16]. The improvement of material’s properties is clear. There are some disadvantages of nanocomposites, such as complex preparation, as particle dispersion is uncontrollable and agglomeration occurs easily, thus restricting composite properties [17].

Recent research has aimed to investigate the role of micro- and nanocomposite preparation in the characteristic microstructure and dielectric properties of materials [18]. Lei et al. summarizes the structural characteristics and basic physical effects of micro–nanocomposites [19]. The properties of different micro–nanocomposites are contrasted; moreover, the development trends of micro–nanocomposites are forecasted. Yin et al. prepared micro- and nano-alumina/epoxy resin with different contents, showing that an addition of 5 wt% nano-alumina particles raised the breakdown strength of epoxy resin by 2% and the addition of micro-alumina particles reduced the breakdown strength of epoxy resin [20]. A study by Roy et al. showed that adding nano-SiO2 particles into XLPE increased the breakdown field strength of crosslinked polyethylene (XLPE) by 20% [21].

The consensus from the literature is that the composite conductance mechanism is closely related to space charge limiting current and ion hopping conductivity, making electrical conductivity characteristics important for composite systems [22]. Therefore, our study explores the electrical conductivity characteristics of micro- and nano-ZnO/LDPE. The LDPE was employed as the matrix and the micro- and nano-ZnO particles were used as fillers. Melt blending is used to prepare the micro- and nano-ZnO/LDPE with different contents. A picoammeter is used to test the electrical conductivity characteristics of pure LDPE and composites samples to explore the composite conductance mechanism. The effects of different composite interfaces and temperatures on the electrical conductivity characteristics of the composites are explored.

## 2. ZnO/LDPE Composites Preparation

The experimental instruments for the cable insulation material preparation and characterisation included a vulcanising press (Fuxin RUbber Co., Ltd., Fuxin, China), a PLM (Aunion Tech Co., Ltd., Shanghai, China), an industrial frequency AC experimental system (Wuhan Moen Intelligent Electric Co., Ltd., Wuhan, China), a vacuum coater (Yuedong Vacuum Equipment Manufacture Co., Ltd., Shantou, China), a differential scanning calorimeter (DSC) (Shanghai Yanjin Scientific Instrument Co., Ltd., Shanghai, China), a node spectrum analyser (Yunfan Equipment Manufacture Co., Ltd., Shenzhen, China), a torque rheometer (Lihang Equipment Manufacture Co., Ltd., Qingzhou, China), and a picoammeter (Damei Equipment Manufacture Co., Ltd., Chongqing, China). The models and manufacturer information of these experimental instruments are presented in Table 1.

The melt blending method is commonly used in composite preparation because of its simple operation and good material mixing. In this study, the micro- and nano-ZnO/LDPE are prepared by melt blending where during the material-mixing, the ZnO powder, LDPE particles, and a small amount of antioxidant are placed into a torque rheometer to prepare the ZnO/LDPE composites. The technological process of preparation is shown in Figure 1.

The specific steps of preparation process are shown as follows:(1)Material Mixing

Firstly, the torque rheometer interior is heated to 140 °C and the electric motor started. All raw materials are added uniformly with 40 g of LDPE used for each sample with 0.12 g of the antioxidant. Samples are made with a mass fraction of nano-ZnO fillers that are 1%, 3%, or 5% of the LDPE quantities. The same is performed for samples with a mass fraction of micro-ZnO fillers of 1%, 3%, 5%, or 10% of the LDPE quantities.

The composites are mixed continuously until the torque curve of the moment and temperature graph begins to straighten (about 20 min). The well-mixed materials are removed gradually and cut into smaller pieces using clean scissors.

(2)Tablet Pressing

The vulcanizing press is preheated for half an hour at 150 °C before the tablet pressing. The polyester membrane surface is wiped clean using a rag with anhydrous ethanol. The mold is placed in the center of two polyester membranes. The metal backing plate is clad outside the polyester membranes. All these are placed into the vulcanizing press and pressurized. When the backing plate touches the upper plate of vulcanizing press, the pressurizing stops. The system is then held for five minutes of preheating, before further pressurizing in stages with several hold times to expel any bubbles from the samples. This produces a smooth sample surface. After the depressurization (approximately 15 min), the templates are removed and left to cool for 10 min before the samples are removed. The samples produced are 0.2 mm thick and the samples’ numbers are marked.

(3)Sample Coating

All the samples are dried. Then an appropriate coating template is chosen. A measuring pole (50 mm diameter) and protective pole are evaporated on one side of the samples. The high-pressure pole is evaporated on a different side. The samples are fastened to the template by tape. The aluminum electrode films are evaporated onto the sample surfaces by vacuum coater. The electrodes can cover opposite sides of the samples, which ensures good electrical conductivity.

(4)Sample Pre-treatment

All the coated samples are taken out from the template. Then these samples are placed in an oven for short circuit treatment at 45 °C for 24 h.

The different samples are numbered in Table 2.

## 3. Materials Structural Characterization and Conductivity Test

The composite materials are characterized to analyze the chemical properties by physical and chemical methods to observe the elemental composition, surface morphology, and crystalline structure. The microstructures of the composites are investigated by the PLM and FTIR methods.

### 3.1. PLM Characterisation

PLM is used to observe the polymer crystalline morphology of pure LDPE and composites. The PLM test results of different samples are shown in Figure 2.

A potassium permanganate and concentrated sulfuric acidic solution is used to corrode the surface of P0, N5 and M5 samples for 5 h. After the corrosion, the samples are removed and cleaned. (Please include an explanation here of what PLM is, what instrument you used with what conditions and settings).

The composites show a tighter crystalline morphology with smaller crystal sizes and more crystals than the pure LDPE, suggesting that adding ZnO particles into the LDPE matrix improves the crystalline structure of LDPE by reducing their size, thus allowing the crystals to arrange more closely. This process reduces the non-crystalline areas between the crystal, thus decreasing the charge carrier’s range motion in the non-crystalline areas. The combined scattering and blocking effect shortens the charge carrier’s mean free path. It is difficult for the carrier to accumulate enough energy to destroy the macromolecular chains.

### 3.2. FTIR Characterisation

The FTIR test identifies the material structure by detecting the transmission and absorption of different frequencies of infrared light. The pure LDPE and different composite samples are tested with FTIR to observe the chemical bond effect of ZnO particles and LDPE matrix when formed into a composite. In this test, each sample is scanned 32 times and then the average value is taken. The FTIR test patterns of different samples are shown in Figure 3.

All three types of sample show the C-H stretching vibration peak at 2840 cm^−1^, the C-H bending vibration peak at 1450 cm^−1^, and the C-H characteristic at 720 cm^−1^. The infrared absorption spectrum intensity of nano-ZnO/LDPE is higher at all frequencies measured than that of micro-ZnO/LDPE. This indicated that the ZnO particles addition did not change the characteristic peak position in LDPE matrix, that is, the ZnO particles did not change the molecular chain structure of LDPE. Around 3500 cm^−1^, the hydroxyl vibration peak of microparticles and nanoparticles almost disappeared because the microparticles and nanoparticles were coated by LDPE macromolecules and were hard to detect in the composites’ preparation. Moreover, the –OH on the ZnO particles’ surface would react with –H to form water molecules during the composites’ preparation. Then, the water molecules would evaporate from the materials.

### 3.3. Conductivity Test

In the conductivity test, the samples are under continuous compression and a non-contact heating method is used. Under the different temperatures, all the samples electrical conductivity characteristics are evaluated and explored.

#### 3.3.1. Experimental Equipment

The testing device of the conduction current is shown in Figure 4. The test system is a three-electrode system that is connected to an adjustable high voltage DC power supply. The different voltages are applied and the current value through the samples monitored.

#### 3.3.2. Experimental Process

The samples and the three-electrode system are placed in a drying oven. Good electrical contact between the electrode and the sample aluminum films is assured. The temperature is held constant throughout the tests while the applied DC voltage is gradually increased. The test temperatures are 25 °C, 45 °C, 60 °C, 75 °C, and 90 °C, respectively. The Picoammeter is used to evaluate the direct current of the samples under different voltages which are 1 kV, 2 kV, 3 kV, 4 kV, 6 kV, 7 kV, and 8 kV. The pressurized initial currents are composed of instantaneous current, relaxation current and conductance current. With increasing time, the current stabilizes to give the conductance current. For consistent results, the system is given 10 min after each increase in voltage to stabilize before the conductance current value is measured. When the testing at one temperature has been completed, the oven temperature is increased to the next chosen temperature and given two hours to stabilize before testing at the new temperature occurs.

(1)Effect of ZnO particles on electrical conductivity characteristics of LDPE

Figure 5 shows the current density vs. electric field strength (J-E) characteristic curve for P0, N1, N3, and N5 samples. In low field regions, the conductance current curve of nanocomposites is indistinguishable from that of pure LDPE. As the electric field intensity increases, the different composites show different electrical conductivity characteristics. The conductance current of N5 is the lowest and the conductance current of N1 is the highest, with N3 in-between, illustrating that the nanoparticles added into the LDPE reduce the conductance current as the electric field intensity increases, with the highest nanoparticle content giving the most significant restriction to the conductance current because the added nano-ZnO particles introduce deep traps and have a scattering effect on the charge carrier.

The steady-state conduction current is formed by carrier movement. For pure LDPE, the carriers are mainly composed of electrons and ions. Under applied voltage, the electron would be injected from electrode to samples. Therefore, as the electric field strength increases, the electric force suffered by the electrons and ions in the dielectric increases. On the other hand, the electrons in the dielectric which are injected from the electrode increases. This causes the accretion of carrier concentration and mobility. The current density of pure LDPE increases. After adding the ZnO particles, the heterogeneous nucleation of ZnO forms the interphase of additive and pure LDPE, from which the traps are introduced and the carrier is trapped. The carrier concentration and mobility decrease. Therefore, the current density of composites is lower than that of pure LDPE. The adding of nano-particles could lead to the deep traps. With the increase of trap density, the nano effect is significant. The current density of nano-composites is the lowest.

Figure 6 shows the J-E characteristic curve of pure LDPE and micro-ZnO/LDPE. In low field regions, the conductance current curve of micro-composites is indistinguishable from that of pure LDPE. As the electric field intensity increases, the conductance current of micro-ZnO/LDPE increases to a greater extent than pure LDPE. The conductance current values order of the samples are as follows: M1 < M3 < M5 < M10, illustrating that the microparticles added into LDPE promote the conductance current with increasing electric field intensity. The higher microparticle content increases the conductance current because the micro-ZnO particles to introduce shallow traps, increasing the micro-composites conductivity.

Figure 7 shows the J-E characteristic curve of samples P0, N5, and M5 to highlight the difference between the micro- and nanocomposites. As the electric field intensity increases, the conductance current of M5 is higher than P0, while the conductance current and conductivity of N5 are significantly lower, illustrating that the same content of particles with different sizes causes contrary effects on the composite conductivity. The addition of the nanoparticles inhibits carrier migration, but the microparticles promote carrier migration.

(2)Samples electrical conductivity characteristics under different temperatures

Figure 8 shows the J-E characteristic curve of samples P0 at different temperatures. The conductivity of sample P0 is unaffected by electric field intensity at room temperature within limits. In the range of field strength tested here (Figure 8), the conductance current of sample P0 increases nonlinearly with increasing field strength. The conductance current increases significantly when the field strength is above 15–17 kV/mm due to field-assisted thermal dissociation occurring in the impurities and antioxidants. The conductivity curve on a log–log scale displays three conspicuous sections. The ohm conductance is the dominant effect in a low electric field. When the electric field is higher than 12 kV/mm, the conductivity curve gradient is greater than in the ohm conductivity regions, implying the presence of an additional non-ohmic conductance. In the range of field strengths measured, the samples follow the Calder law with a step change between the space charge limited current when trapped and the region with filled traps clearly displayed in the room temperature tests. The conductivity curve of samples P0 changes significantly with a change in temperature. At the same field strength, the conductance current of samples P0 changes with the increasing temperature. When the temperature increases to 60 °C and above, the distinct three-section dependence of conductance current on electric field intensity is lost and the current at each temperature at high electric field intensity rises to a limiting value.

The J-E characteristic curves of samples N1, N3, and N5 under different temperatures are shown in Figure 9. The conductance current density rises with the temperature increasing from 25 °C to 90 °C. At high temperatures, the current density curve of nanocomposites with changing field strength do not show the distinct trilinear shape, suggesting that the nanocomposite conductance current is suppressed by space charges and is affected by other factors such as ion hopping conductivity. At elevated temperatures, the conductivity of low content nanocomposites shows little change with increasing electric field up to a certain value, illustrating that the dependence of conductivity on electric field has decreased in this situation.

The J-E characteristic curves of M1, M3, M5, and M10 samples under different temperatures are shown in Figure 10. The curves between 25–90 °C show that the conductance current density rises with increasing temperature. At high temperatures, the conductance current density curve of micro-composites with changing field strength do not show the distinct trilinear shape illustrating as in the nanocomposite case that factors other than space change suppression affect the conductance. At high temperatures, the conductance current and conductivity are minimally affected by increasing the electric field, illustrating the decreased dependence of conductivity on electric field in this situation.

The conductivity–temperature characteristic curve of samples P0, N5, and M5 under the 9 kV/mm field strength is shown in Figure 11. The conductance current increases with increasing temperature in all samples. The conductivity of the M5 samples rise quickly compared with that of P0 over the same temperature change while the conductivity of the N5 samples show less of an increase. Below 60 °C, the conductance current of all composites is almost independent of temperature and the conductivity–temperature characteristic curve of the three samples are indiscernible from one another. At 60 °C and above, the conductivity of M5 and P0 samples change significantly with the temperature rise when the N5 samples remain almost unchanged because the charge carrier concentration increases with temperatures above 60 °C. As the charge carrier migration rate is linked to conductivity increases, the conductivity increase in P0 and M5 samples is higher than in N5 samples because the nanoparticles added into the LDPE matrix create a close interface structure, which hinders the carrier’s migration.

Within the test range of the whole experimental temperature, the current density of different samples increases with the rises in temperature. While the temperature is low, the conductivity changes slightly. At this time, the current density of the samples is less dependent on temperature. Under the same temperature, the current density of nano-composite is lower, and micro-composite is higher. This is due to the temperature rise, the carrier in the nano-composite takes more power and overcomes the barrier between energy bands. The carrier concentration and current density increases. On the other hand, when the temperature is constant, the temperature mainly affects the ionic conductance in the dielectric. The ionic conductance could be divided into intrinsic ionic conductance and weakly bound impurity ions conductance. At low temperatures, the conductance in the dielectric is formed by weakly bound impurity ions. While at high temperatures, the polymer disassociation would lead to the emergence of intrinsic ionic conductance. The carrier concentration increases further, and the varying degrees of current density with temperature is greater. However, with micro-ZnO particle addition, the shallow traps would be introduced into the matrix. The intermolecular force between micro-particles and polythene matrix is weak. With the rise in temperature, the carrier migrates easily. Therefore, the conductivity of micro-ZnO/LDPE composites is affected more by the temperature. The nano-particle addition would introduce deep traps into the polythene matrix, which effectively restrains the carrier migration. Therefore, the conductivity of nano-composites is less affected by temperature.

## 4. Conclusions

In this study, we focused on the electrical properties of pure LDPE, nano-ZnO/LDPE, and micro-ZnO/LDPE. PLM and FTIR tests highlighted the differences in the samples’ crystalline structure and chemical bonding. The conductivity of these samples is measured under different thermal and electrical situations to obtain the electrical conductivity characteristics of pure LDPE, nano-ZnO/LDPE, and micro-ZnO/LDPE. The specific conclusions are as follows:(1)With increasing electric field intensity, the addition of nanoparticles into LDPE causes an inhibiting effect on the conductance current. The inhibition effect is more prominent with higher filler content because the nanoparticles added into the LDPE introduce deep traps and scatter charge carriers. The micro-particles increase the conductance current of LDPE in proportion to the filler content.(2)At normal temperatures, the LDPE results agree with the space charge limited current (SCLC) theory with the clear ohmic conductance regions and the non-ohmic conductance regions present in the J-E curve. As the temperature increases, the gradient of the LDPE J-E characteristic curve reduces, and the trilinear shape is no longer visible. When the temperature is at 60 °C and above, the increase of the electric field intensity to a certain value shows a reduction in the dependence conductance current and conductivity on the electric field intensity. The nanocomposites display a similar trend with temperature as the micro-composites. The inhibition effect of 5 wt.% nano-ZnO/LDPE on charge carriers’ mobility is the highest and the promotional effect of 10 wt.% micro-ZnO/LDPE on carriers’ mobility is the highest.(3)With increasing temperature, the conductivity of pure LDPE, micro-ZnO/LDPE, and nano-ZnO/LDPE increase by differing amounts. Charge carrier traps are introduced into micro-ZnO/LDPE compared with pure LDPE leading to a greater increase in conductivity with temperature. The interface structure in the nano-ZnO/LDPE samples causes the conductivity to remain low with increasing temperature.

## Figures and Tables

**Figure 1 molecules-27-03674-f001:**
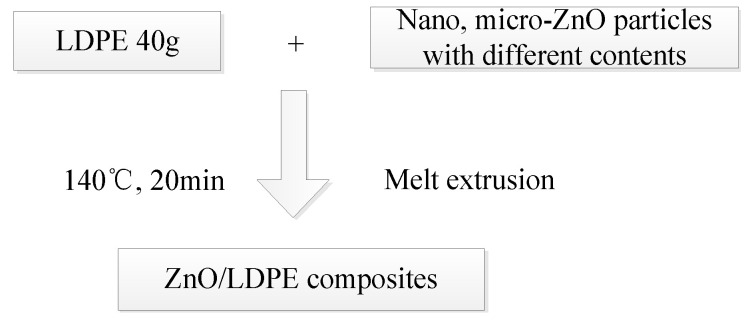
Preparation of micro and nanocomposites.

**Figure 2 molecules-27-03674-f002:**
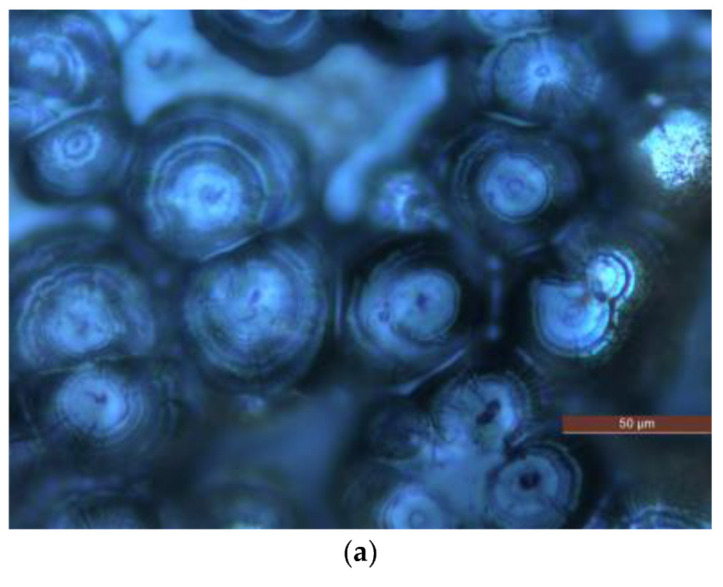
PLM patterns of samples P0, N5 and M5. (**a**) Crystalline morphology of sample P0. (**b**) Crystalline morphology of sample N5. (**c**) Crystalline morphology of sample M5.

**Figure 3 molecules-27-03674-f003:**
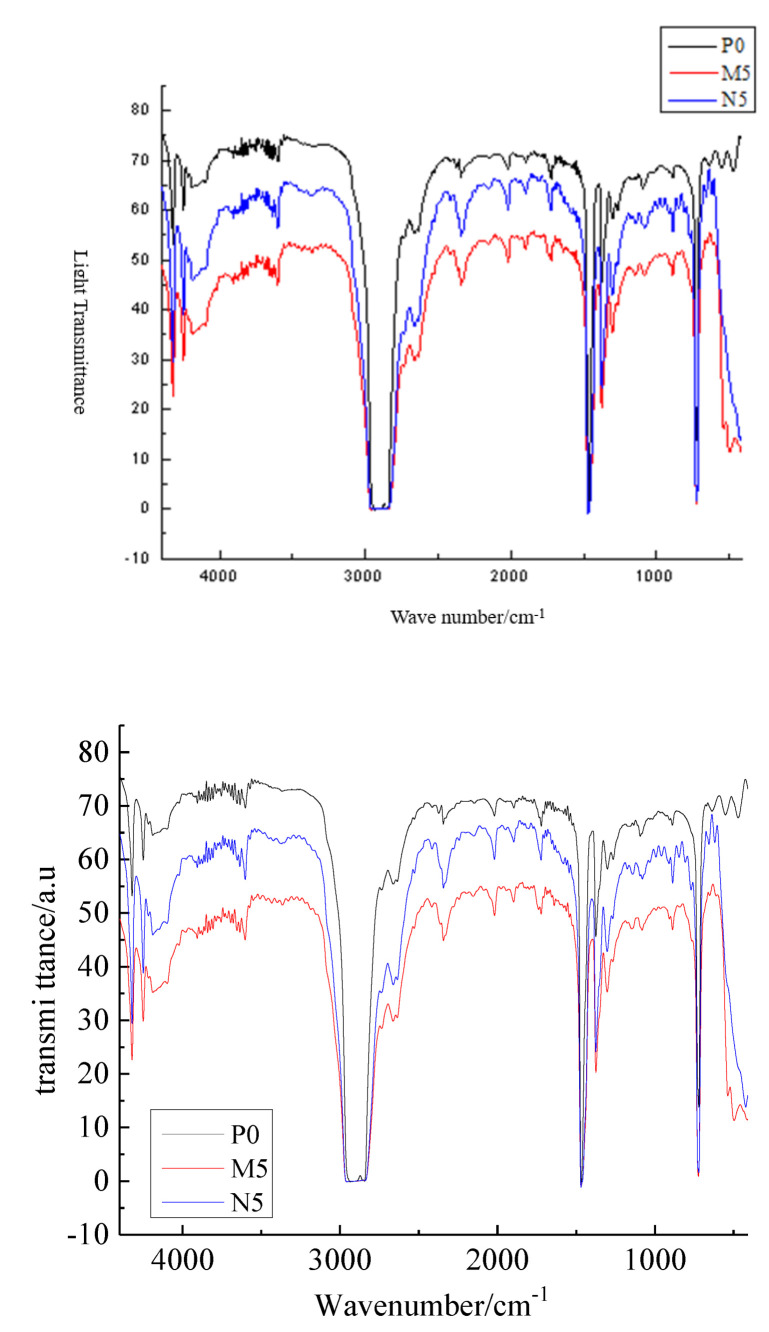
FTIR pattern of different samples.

**Figure 4 molecules-27-03674-f004:**
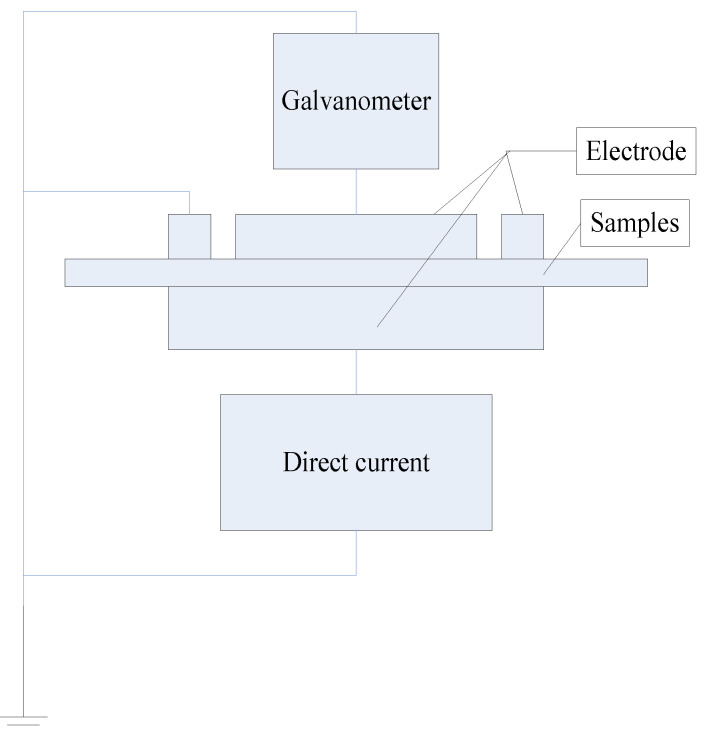
Conductivity characteristics testing device.

**Figure 5 molecules-27-03674-f005:**
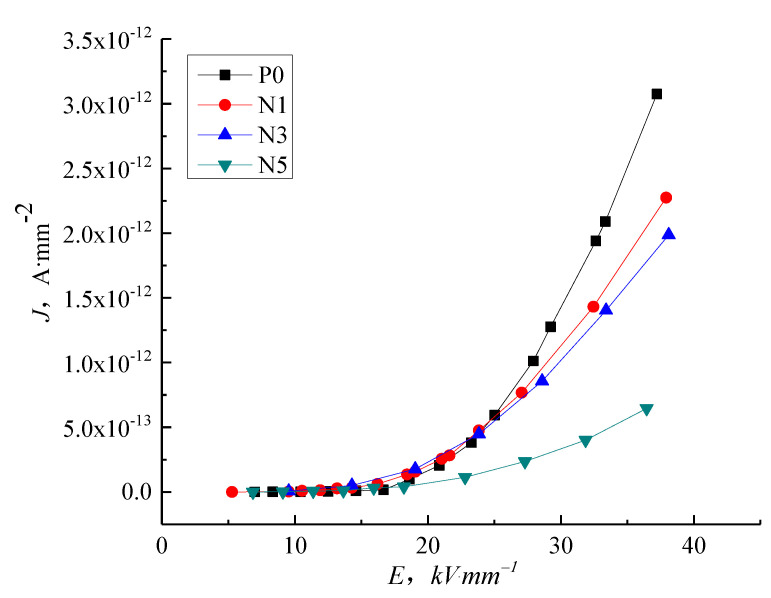
J-E characteristic curve of LDPE and nano-ZnO/LDPE with different content.

**Figure 6 molecules-27-03674-f006:**
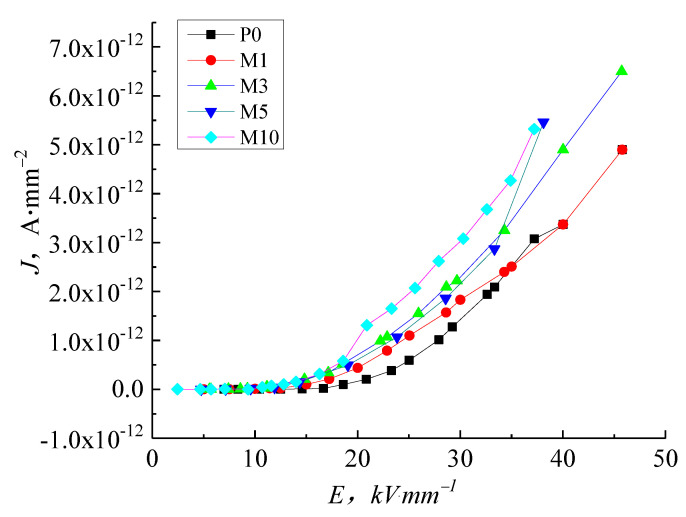
J-E characteristic curve of LDPE and micro-ZnO/LDPE with different content.

**Figure 7 molecules-27-03674-f007:**
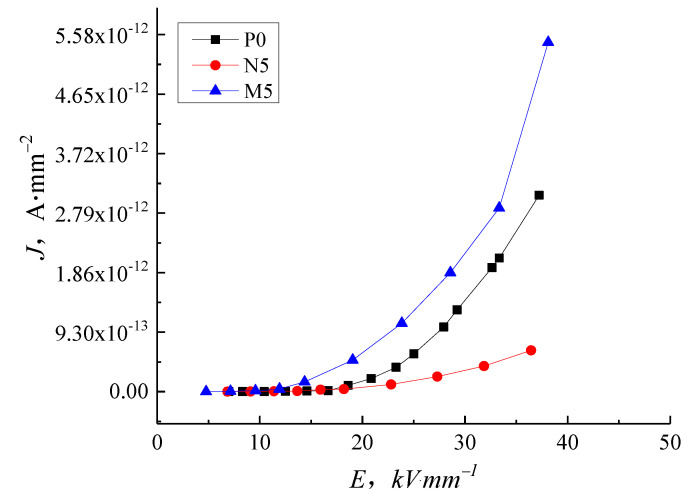
J-E characteristic curve of LDPE, nano-ZnO/LDPE and micro-ZnO/LDPE.

**Figure 8 molecules-27-03674-f008:**
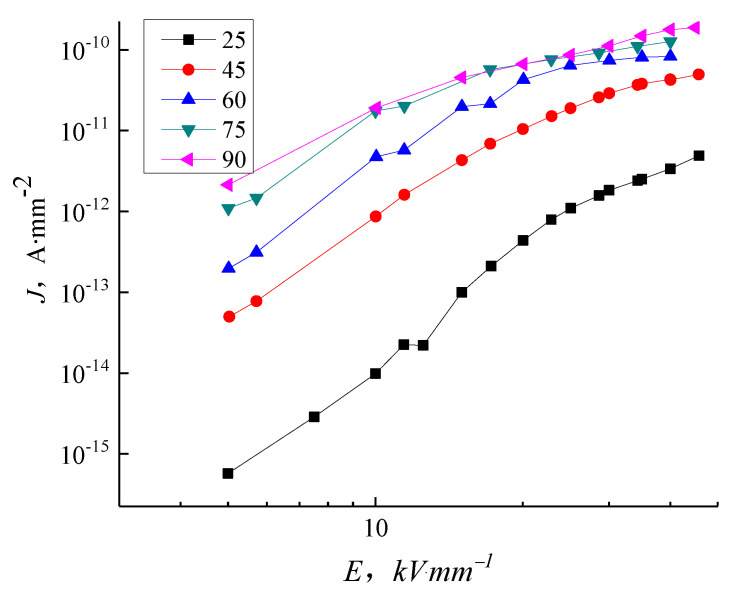
J-E characteristic curve of LDPE in different temperatures.

**Figure 9 molecules-27-03674-f009:**
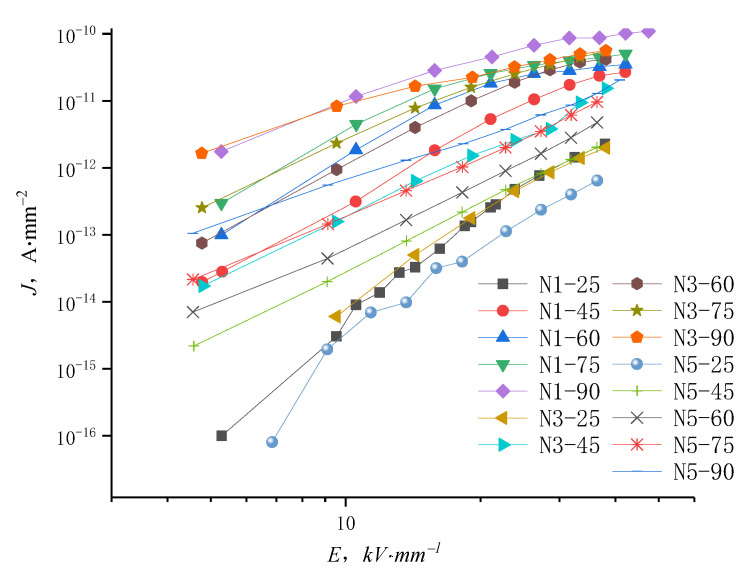
J-E characteristic curves of nano-ZnO/LDPE in different temperatures.

**Figure 10 molecules-27-03674-f010:**
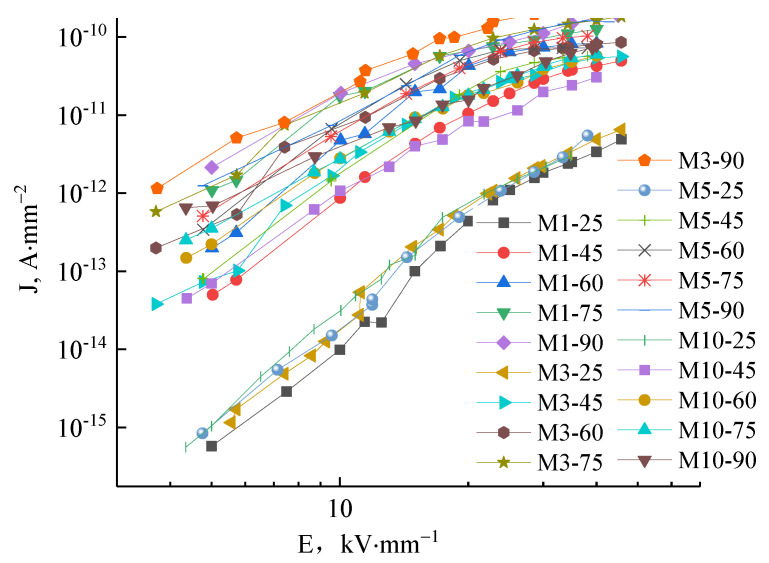
The J-E characteristic curves of M1, M3, M5 and M10 in different temperatures.

**Figure 11 molecules-27-03674-f011:**
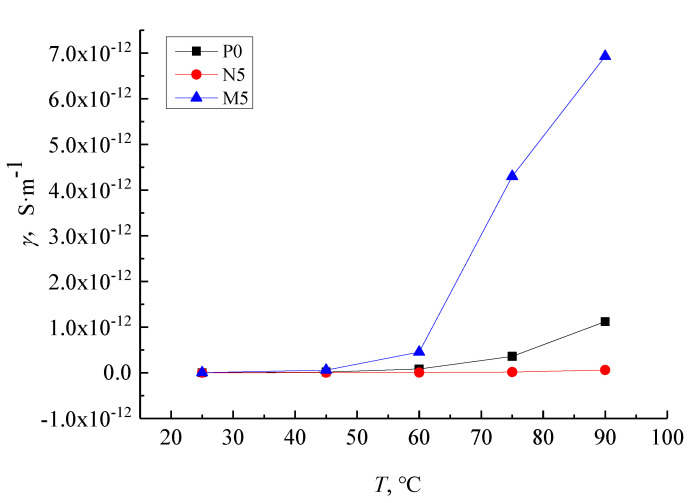
The J-E characteristic curves of LDPE, nano-ZnO/LDPE and micro-ZnO/LDPE in 9 kV/mm.

**Table 1 molecules-27-03674-t001:** Experimental equipment used in this study.

Equipment Name	Model	Manufacturer
Electronic balance	YP202N	Sincere Dedication of Science and Technology Innovation, Shanghai, China
Electric booster mixer	JJ-1	Rong Hua Equipment Manufacture Co., Ltd., Suzhou, China
Constant-temperature water-dissolving pot	DSY-2-4	Shu Li Instrument Manufacturing Co., Ltd., Shanghai, China
Adjustable electric jacket	SHSL	Shu Li Instrument Manufacturing Co., Ltd., Shanghai, China
Vacuum pump	2XZ-1	Huang Yan Vacuum Pump Factory, Ningbo, China
Vacuum drying oven	DZF-6020MBE	Boxun Industry and Commerce Co., Ltd., Shanghai, China
Sand core funnel	G4	GuangDa Glass Co., Ltd., Changchun, China

**Table 2 molecules-27-03674-t002:** Number and material composition of different samples.

Samples Number	M-ZnO	N-ZnO
P0	0 wt%	0 wt%
N1	0 wt%	1 wt%
N3	0 wt%	3 wt%
N5	0 wt%	5 wt%
M1	1 wt%	0 wt%
M3	3 wt%	0 wt%
M5	5 wt%	0 wt%
M10	10 wt%	0 wt%

## Data Availability

Not applicable.

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
