# Peer review of "Study of the Electrical Conductivity Characteristics of Micro and Nano-ZnO/LDPE Composites"

_molecules, 2022, doi:10.3390/molecules27123674_

Round 1

Reviewer 1 Report

The authors discuss the electrical conductivity characteristics of micro and nano-ZnO/LDPE composites, with relevance to the effects of temperature on the characteristic behavior of J-E. I consider the choice of topic current and relevant for potential industrial applications. The introduction is well written and clearly identifies the research objectives. The results are explained in detail in section 4, which makes no sense for this information to appear after the discussion. This should be moved to the beginning and placed right after the introduction. Otherwise, the content of the article cannot be understood, as it is only at the end that the characteristics of the samples are identified.

I also have the following comments:

Abstract: insert Polarized Light Microscopy (PLM)

Introduction: define XLPE

FTIR characterization: transmission instead of reflection

Figures 7 and 8. Insert should include 25°C and so on

Please remove [J] from some references. Unless it means something

I recommend major revision

Reviewer 2 Report

In the manuscript, the authors reported how the embedding of micro/nano ZnO nanoparticles affect the conductivity at elevated temperatures and electronic field strengths. After going through the whole manuscript, I would recommend this manuscript to be accepted for publication after revision, the authors should address the following comments.

  1. The figure captions are not clear, the authors are suggested to provide captions with more details to enhance the readability. 
  2. There are too many J-E curves, the authors are strongly suggested to combine some of them into in figure for easy comparison and readability instead of including them as single figures.
  3. The discussion section is missing, the authors seemed to copy the previous review comments wrongly to this section. 
  4. Comprehensive and insightful discussion is needed, otherwise the manuscript would be just like an experimental report.

Round 2

Reviewer 1 Report

In view of the previous requested revision, I am satisfied with the changes made by the authors. The manuscript is now ready to be published. I don't have anything else to add.

Author Response

Thank you for your approval.

Reviewer 2 Report

In the manuscript, the authors made some revision, however, the quality of the manuscript can be further improved.

  1. There are still too many figures, the authors are suggested to include those related J-E figures as separated panels.
  2. The captions of the figures are still not clear, the authors should add more details to facilitate the readers to understand the figure easily. 

Author Response

Thank you for you suggestion. I have included the related J-E figures. And the captions of the figures are imporoved.